# Embedded in Nature: Challenges to Sustainability in Communities of the Greater Yellowstone Ecosystem

**Ryan D. Bergstrom** [1,*]  **and Lisa M.B. Harrington** [2]

1   Program in Geography, University of Minnesota Duluth, Duluth, MN 55812, USA
2   Department of Geography, Kansas State University, Manhattan, KS 66506, USA; lbutlerh@ksu.edu
*   Correspondence: rbergstr@d.umn.edu; Tel.: +1-218-726-6620

**Abstract:** Solutions to sustainability transitions tend to be applicable for specific regions but not the whole of society. Limitations on what may be sustained also exist, and preferences will vary among people in different places. Because of these differences, there is a need for better understanding of the perceptions and experiences of local community members and the challenges they face in the transition toward sustainability to promote realistic and effective decision-making. As a region with significant natural resource protections, the Greater Yellowstone Ecosystem has been known to researchers for decades as an ideal location to study human-environment interactions. The objective of this study was to determine the challenges to sustainable community development and natural resource management identified by residents of communities surrounding Yellowstone and Grand Teton National Parks. Thirty-two key informant interviews were conducted with decision-makers, with a focus on the communities of Red Lodge and West Yellowstone, Montana, and Jackson, Wyoming. Findings suggest that primary challenges include the seasonality of the tourist industry, disparities between agricultural and tourism-dependent priorities, and the implementation of stated sustainability goals. Challenges differ based on communities' socio-economic conditions, dependence on tourism and recreation-based industries, and the influence of local and extra-local institutions.

**Keywords:** sustainability; sustainability transition; Greater Yellowstone Ecosystem; community development; economic development; natural resource management

## 1. Introduction

The American West has been in the midst of a fundamental transition for several decades. Parts of the region have shifted to a service-based economy with low-skilled tourism and recreation-based jobs, and high-tech/professional industries based on telecommuters and "lone eagles" [1]. Catalysts for these changes have included declining commodity prices and a globalized economy; the consequences have been sometimes profound shifts in local economies and socio-cultural characteristics [2]. It is unclear whether the new economic foci will be sustainable over the long-term; we must be cognizant of the challenges we face on our transition toward sustainability, including the potential effects of change and our ability to mitigate and cope with change [3–6]. The question of whether societal and economic needs can be met while simultaneously maintaining the planet's life support systems needs to be addressed at local and regional, as well as global, scales.

In order to successfully transition toward sustainability, a better understanding of coupled human and natural systems is critical. Because of the close couplings between these systems in the Greater Yellowstone Ecosystem (GYE) (Figure 1), it is an ideal location to study these relationships. While sustainability visions, goals, and objectives may be similar across a region, consideration of local contexts affecting perceptions provides valuable understandings that may inform sustainability

pathways at local scales. The local/regional knowledge gained here—and the approach used to gain it—may provide insights contributing to global sustainability.

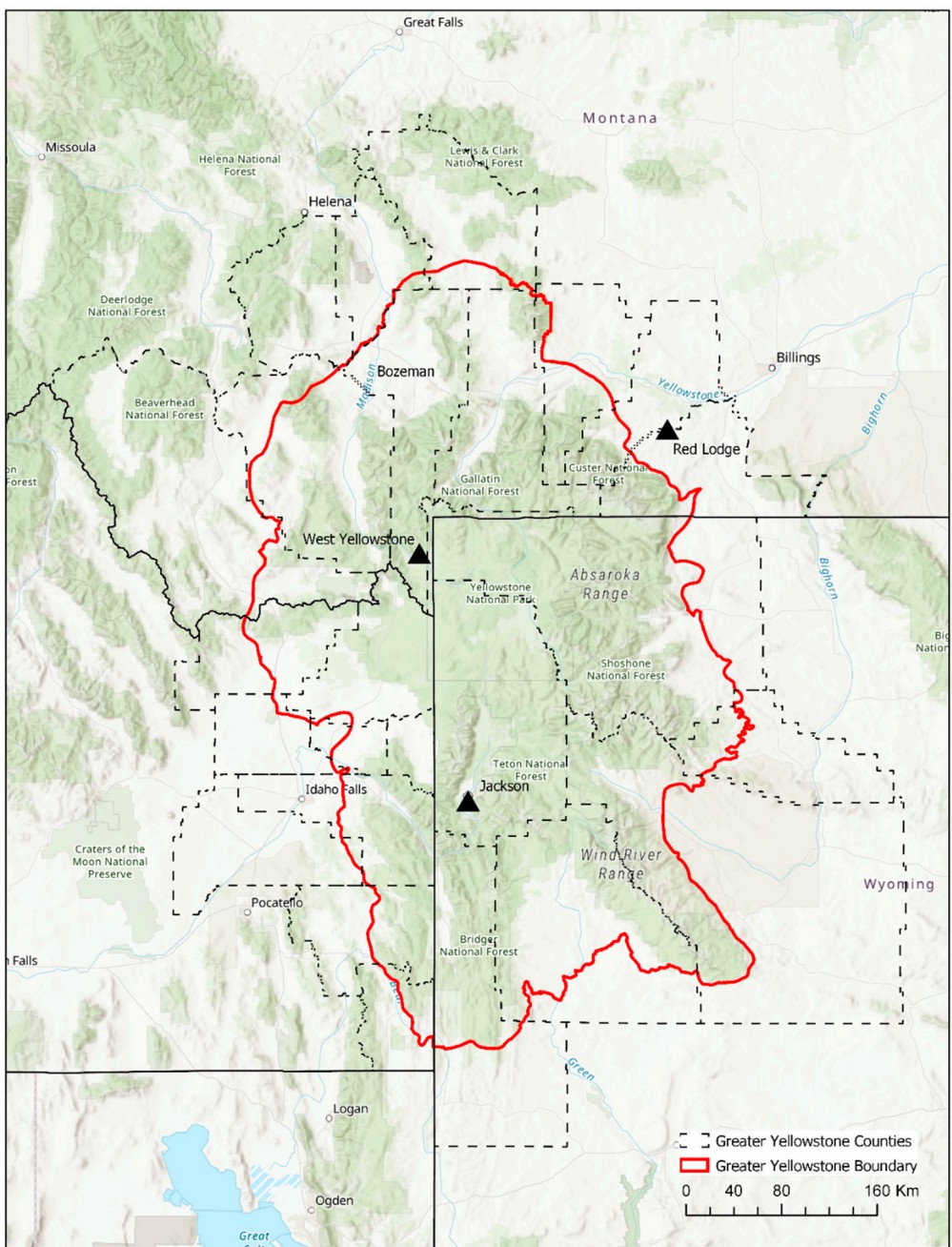

**Figure 1.** The Greater Yellowstone Ecosystem.

The Greater Yellowstone Ecosystem (GYE)—Yellowstone and Grand Teton national parks, and the natural and built environments of adjacent areas—is a prime example of a region that has witnessed a prolonged shift to a more service-based economy [7]. Because of the coupled and complex nature of social and ecological systems in the region, with a combination of natural resource extraction, increasing tourism and recreation-based industries, lack of affordable housing, and an influx of amenity-driven migrants, decision makers in the region face new and increasingly complex challenges in managing the region holistically [8,9]. For example, a local leader in Jackson, Wyoming, a town within the GYE, retold a story about a resident who claimed he "didn't give a damn" about affordable housing in the community, provided he could get a table at an upscale restaurant in town. The politician

had to calmly explain to the resident that without affordable housing he would likely be waiting on himself [10]. This example highlights one of the many challenges that communities in the GYE face on their transition toward sustainability: challenges often stem from perceptions, behaviors, and values of the stakeholders and decision makers. Such stakeholder perceptions and values are additional to the perhaps broader understanding that challenges can be systematic and bureaucratic in nature. In the case of the GYE, these include changes to winter access to Yellowstone National Park (YNP) and the management of bison who wander outside the park.

The objective of this study was to build understanding of the challenges three communities in the GYE face on their transition toward sustainability, and to relate those challenges to conceptualizations of sustainability [3]. Communities, for the purpose of this study, are defined as cities or towns. While many of the challenges to communities and environmental conditions in the GYE are well known, it is critical to have a clear idea of local residents' perceptions of the most important difficulties facing long-term sustainability in order to take effective action. Specifically, we wanted to determine the perceptions of decision makers and other stakeholders regarding local and regional challenges to transitions toward sustainability. Understanding of stakeholder perceptions is important because it provides critical insights into the socio-cultural dimension of coupled and complex socio-ecological systems [11]. Findings suggest that challenges include the seasonality of the tourist industry, disparities between agricultural priorities at the county level and tourism-dependent community priorities, and the implementation of sustainability goals and priorities. Challenges differ based on communities' socio-economic conditions, dependence on tourism and recreation-based industries, and the influence of local and extra-local institutions.

## 1.1. The Greater Yellowstone Ecosystem

The Greater Yellowstone Ecosystem is an ideal location to more fully examine the challenges that communities face on their transitions toward sustainability for a variety of reasons. First, the GYE is physically, socially, and administratively diverse, covering over 20 million acres in 20 counties and parts of three states. Second, the GYE is one of the fastest growing regions of the country [12], which creates challenges to sustaining desirable conditions. For example, Bozeman, Montana, located just 80 miles from the western entrance to YNP, is one of the fastest growing cities in the nation, with an annual growth rate of 4.3 percent [13]. If growth rates of the past 30 years continue, it is anticipated that the GYE will grow from its current population of 450,000 to 677,000 in the next 13 years [14]. In addition, changes to land uses have had significant impacts on the region's social-ecological systems. Changes have involved extraction of natural resources, loss of agricultural lands, residential development [15], and increased tourist visitation rates [16]. Examination of these issues reveals stakeholders' diverse frames of understanding, as well as the complex interrelations among interest groups, government agencies, and individuals in the region.

The GYE faces a number of well-known and long-term challenges to transitioning toward sustainability, most of which are based on anthropogenic disturbances via natural resource extraction activities that are particularly harmful to ecosystem services [17]. As Swanson [18] suggested, the sustainability of local and regional economies that are dependent on oil, gas, and minerals requires land managers, industry representatives, and local communities to ensure that current and future development be conducted in ways that protect environmental resources. The region also faces both environmental and social challenges related to mining [19–21], hydrothermal energy development [22,23], agriculture [24–26], rapid population growth [14], development and zoning policies [15,27], changing ranchland ownership and management [28], the multiple and sometimes conflicting mandates of management agencies [29]; and increasing pressures from tourism and recreation-based visitation [28,30–33]. Driving a number of these changes, amenity migration affects both the environment and social relations, and is itself largely driven by the attraction of environmental beauty and open space [25,28,30,34,35]. A number of highly contentious, highly polarizing, and highly publicized challenges, sometimes termed "wicked problems" [36,37] also face the GYE. Wicked

problems, concerns with no clear solution that pit a myriad of stakeholders with differing frames of understanding against each other, include the reintroduction of the gray wolf and its delisting from the endangered species list [38], on-going modifications in winter access to Yellowstone National Park [39], bison management and its relationship to brucellosis [40], and proposed mineral and energy development [18]. Complicating these narratives is the fact that policy and decision-making for these issues occurs at multiple scales (local to national), with a variety of agencies and institutions involved in promoting or hindering a sustainability transition [41].

### 1.2. A Framework for Sustainable Community Development

For the purposes of this study, sustainability is seen as the ability to reconcile social and economic goals within the limits of the natural environment [7]. The National Academy of Sciences' Board on Sustainable Development [42], went a step further, defining a transition toward sustainability as the ability to meet societal needs by "moving away from actions that degrade the planet's life support systems and living resources, while moving forward toward those that sustain and restore these systems and resources." Thought of another way, sustainability is ultimately a pursuit to improve, or at the very least maintain, desirable societal and environmental conditions over the long term [3].

As Kates [5] suggested, sustainability studies attempt to improve understanding of complex human-environmental interactions and seek to find solutions to the issues facing a sustainability transition, with the ultimate goal of creating knowledge and moving that knowledge into action. Because the values, attitudes, and behavior of people affect actions and priorities, identifying and analyzing these traits, as well as the subsequent actions of groups or individuals, aids understanding and decision-making in the service of sustainability. Key considerations include values related to human use of the Earth; valuation of both social and natural capital in economic terms; and understanding of how individual barriers (e.g., lack of time, money, and knowledge) and structural barriers (e.g., laws, regulations, social norms) promote or hinder successful transitions [43].

Sustainability transitions are geographic processes, and while traditionally this aspect has been underplayed [44], there is growing attention given to the importance of local context [11] and sustainable community development. Sustainable community development has become one of the more important [45] and debated concepts related to sustainability in the past decade [46–48]. Complicating the discourse on sustainable communities is the fact that social sustainability, as one of the three fundamental dimensions of sustainability, is poorly understood and rarely examined in isolation [46,49–51].

The conceptual framework for sustainable community development is based on three dimensions of sustainable development (economic, environmental, societal), also known as the triple bottom line, the three pillars, or the 3P's (people, planet and profit) [45,52,53]. However, it also includes defining characteristics such as social justice via citizen empowerment, ecological stewardship, and economic self-reliance via diversification [54,55]. For Roseland and Spiliotopoulou [56], a sustainable community adheres to traditional definitions of sustainability by being able to meet the social and economic needs of its residents, while protecting the natural environment that supports it. Bridger and Luloff [45] took this a step further by suggesting that, not only does a balance between environmental and development objectives need to be achieved, but a sustainable community promotes "humane local societies." The idea of promoting human well-being was also supported by Storey [49], who further suggested that such support was predicated upon a set of common goals and values within a given community. Lastly, Dempsey et al. [46] noted that sustainable communities include strong senses of place based on community cohesion, trust, and a sense of pride in the community that provide stability, but which also make the community resilient to rapid changes [46,57]. Unfortunately, the changes seen with amenity migration also can be seen as reducing traditional senses of place [58,59].

Local sustainability concerns are increasingly relevant as the magnitude, frequency, and impact of human production and consumption activities are increasingly focused at the community level, and because interventions through policy and decision making are most pronounced, and thus beneficial,

at the local level [45]. While national and international agencies and institutions may play a role in successfully transitioning toward sustainability, in reality it is at the community level that tangible changes are taking place though citizen-led, grassroots efforts [60]. Even though cumulative problems affect broader scales, it is logical to address sustainability concerns at the local level. With that said, it remains unclear how local communities are indeed moving toward a more sustainable future [61].

Sustainable community development can be thought of as relating to community resilience [62]. Emerging as a framework for better understanding community development in the late 1980s, and foci of public interest since the 2000s, resiliency and sustainability have been considered interrelated perspectives for some [63], and independent concepts for others [64]. For Magis [65], community resilience is achieved by thriving amidst change, and thus, resiliency is an indicator of social sustainability. Ultimately, sustainable community development is about finding a middle ground where community resilience is achieved through investments in five capitals (human, social, built, natural, and economic) [56,57]. Arising in the 1990s, the concept of multiple capitals [66] is based on the idea that, in order to be sustainable, it is not possible to build one set of capital stocks at the expense of another [67]. Thus, to be sustainable, communities must mobilize citizens to strengthen their community capital(s) [56], while being staunch stewards of natural capital stocks. Human capital includes the skills and knowledge necessary for community members to maintain an acceptable standard of living, while social capital relates to the relationships and trust built among community members that helps facilitate cooperative decision making. Built capital includes the physical structures and networks such as roads, communications, and buildings, and economic capital is the financial resources of the community. Lastly, natural capital includes the natural resources and ecosystem services provided by the natural and modified environment [46,57,67].

Further, because the challenges related to a successful transition toward sustainability vary in space and time, there is a need for improved understanding of the trends and driving forces of change (challenges) at local levels [68]. In an attempt to determine the driving forces of tropical deforestation, Geist and Lambin [69] differentiated between proximate, or direct, and underlying drivers of change. Proximate drivers require immediate action and often involve human activities at the local level. Examples include agricultural expansion, timber extraction, and development. Underlying driving forces are social processes that underlie proximate drivers and, while they may operate at the local level, their impact is often experienced at larger spatial scales. Examples include demographic change, economic and institutional characteristics, and policy decisions. In the American West, proximate drivers include the conversion of agricultural lands to subdivisions, mineral extraction [16], and tourism and recreation activities that degrade the natural environment. Underlying drivers include rapid population growth during the last three decades [15], federal land use policies [70], and a lack of uniform planning and zoning policies [71].

Lastly, the traditional conceptual model of the triple bottom line first suggested by the Brundtland Report [72] represents sustainability dimensions (economic, environment, society) as separate and independent systems. In this model, sustainability dimensions rarely interact, although when interaction and integration occur (i.e., attempts to balance the triple bottom line), system sustainability may be achieved. The overemphasis or lack of focus on any single dimension results in an unsustainable system. In contrast, it has been argued [3,73] that sustainability dimensions do not exist as separate systems, but rather that they are intricately embedded in a coupled and complex nature-society or human-environment system where no single dimension can adequately function without the others. In this way, the economy is embedded in and dependent upon society, and society is embedded and dependent upon the goods and services provided by the environment. However, the embedded nature of sustainability suggests there will be states of differing levels of sustainability (from lesser to greater) for any location, material, or condition, and thus not only does place matter, but also choices for and by individuals, communities, and society [3]. As Bridger and Luloff [45] suggested, each community's or region's journey toward becoming sustainable will differ because each community's

contexts differ, including their "environmental problems, natural and human resource endowments, levels of economic and social development, and physical and climatic conditions".

## 2. Methods

In order to explore residents' most pressing concerns regarding the pursuit of sustainability, this project's primary data were gathered from three selected communities in the GYE. The decision to conduct comparative community studies is based on the idea that, as key linkages between the natural world and humans, community conditions can provide insights into environmental and societal processes and decision-making [74], as well as community resiliency [65]. Communities were chosen based on a combination of identification as national park "gateways", making them desirable locations for tourism, recreation activities, and amenity migration; and differing socio-economic and natural characteristics. Three locations in Montana and Wyoming—West Yellowstone, Red Lodge, and Jackson—were selected (Table 1). Gateway communities include the cities and towns that are adjacent to public lands that draw tourists, recreationists, and migrants due to their high quality of life and scenic beauty [75]. In the GYE, Yellowstone and Grand Teton National Parks are the core natural areas attracting visitors, but other public lands like National Forests are also very important. The national park areas are managed by the National Park Service (NPS) and the national forests are managed by the US Forest Service (USFS); other public lands also make up parts of the region.

**Table 1.** Socio-economic characteristics of study communities: 2012–2016 [76].

| | Population | Agriculture, Forestry, Fishing, Hunting, and Mining (% of all workers) | Retail (%) | Professional, Scientific & Management (%) | Education & Health Care (%) | Entertainment, Accommodation & Food Services (%) | Median Income | Median Home Value | Below Poverty Line (%) |
|---|---|---|---|---|---|---|---|---|---|
| West Yellowstone, Montana | 1139 | 1.9 | 14.3 | 3.7 | 4.6 | 46.9 | $32,134 | $244,400 | 14 |
| Red Lodge, Montana | 2328 | 4.6 | 5.6 | 11.2 | 20.7 | 19.6 | $42,120 | $227,900 | 19.3 |
| Jackson, Wyoming | 10,279 | 3.1 | 9.8 | 9.5 | 16.2 | 33.3 | $70,517 | $573,400 | 8.8 |

West Yellowstone, located in Gallatin County, is the smallest of the study communities. Although its population is under 1500, it is one of the most-visited communities by tourists traveling to Yellowstone National Park. Of the four million tourists who visited the park in 2017, 42 percent (nearly 1.7 million), entered through West Yellowstone [16]. West Yellowstone has experienced 22 percent growth since 2000, with a growth rate of more than 3 percent per year since 2010. The community is expected to grow an additional 26 percent by 2037 [77]. Population increases will have a substantial impact on housing availability and costs, especially because the town is physically constrained on all sides by public lands, with median household income 20 percent lower than that of the entire state, and nearly 40 percent lower than that of Gallatin County as a whole [77].

Red Lodge is also very small, with a population of less than 2500. It is located in Carbon County, approximately 70 miles from Yellowstone NP, and has a much wider array of employment than found in West Yellowstone. That said, tourism and recreation services still constitute a significant portion of the community's economy, and the closure of the Beartooth Highway during the winter months, one of the largest tourist draws, results in a seasonal economy [78]. Lastly, the community experienced very little growth in the 2000s, and projections for Carbon County suggest that it will continue to see slow or declining growth through 2060 [78].

Jackson is the gateway to Grand Teton National Park, which is connected to Yellowstone National Park via the John D. Rockefeller, Jr. Memorial Parkway. With a population of over 10,000, the community is large enough to support more stand-alone industries than Red Lodge and West

Yellowstone, although over 30 percent of residents are employed in service-related industries. One of the largest differences between Jackson and the other two communities is its tremendous growth in recent decades as a result of amenity migration. Home values are some of the highest in the country, exacerbated by relatively little land in private ownership. The amenity growth and second home boom are reflected in the community's higher median income ($70,517) and its very high median home value ($573,400) [76]. Such statistics obscure income disparities, however, with many residents having relatively low incomes as compared to the very wealthy. Highlighting the extreme economic disparities in the community, Jackson was named the most economically unequal city in the country in 2016, with the bottom 90 percent of earners only accounting for 17.3 percent of income [79]. According to the Jackson Hole Community Housing Trust [80], this inequity causes substantial challenges for service-related workers. These challenges stem from the fact that only 11 percent of the total housing stock in the county is considered affordable; as a result, 38 percent of employees are forced to live outside the county, most often commuting from western Idaho across the 2570 m Teton Pass. The 18 km travel route is particularly dangerous in winter months due to its elevation.

*Key Informant Interviews*

The key informant interview technique [81] allows for an improved understanding of the opinions and beliefs of informants who may have specialized knowledge based on their positions in society [82,83]. This approach is particularly well suited to studying the patterns of problems and their causes within societies, and allows those patterns to be linked to local, regional, national, and global socio-economic and socio-political structures. Patterns and connections may then be utilized to provide beneficial plans of action for local communities [84].

A list of potential key informants was developed using a purposeful sampling approach [82] to increase the likelihood that informants would possess a working knowledge of the concept of sustainability and the challenges their communities face. This list was compiled using local Chambers of Commerce, city and county offices, and local, regional, and national non-government organizations (NGOs). Key informants were initially contacted via email or mailed letters to explain the project and inquire about availability for interviews. In addition to scheduled interviews, snowball interviews were conducted based on input from key informants. Three interviews were conducted via email based on informant preference and availability. Eight to 10 interviews were conducted in each of the study areas, for a total of 32 key informant interviews. The proportions of interviews from each category of informant (government, business owner, non-government organization) varied due to the availability of interviewees (Table 2).

**Table 2.** Key Informant Demographic Data.

| Location | Business Owner | City Official | NGO | Average Years in Residence | Male | Female | Total |
|---|---|---|---|---|---|---|---|
| West Yellowstone, MT | 6 | 2 | 2 | 34 | 7 | 3 | 10 |
| Red Lodge, MT | 2 | 4 | 4 | 19 | 6 | 4 | 10 |
| Jackson, WY | 0 | 7 | 2 | 16 | 8 | 1 | 9 |
| Regional NGO | 0 | 0 | 3 | 18 | 1 | 2 | 3 |
| Total | 8 | 13 | 11 | 22 | 22 | 10 | 32 |

The semi-structured interview questionnaire consisted of 11 open-ended focal questions and two open-ended probing sub-questions that were approved by the Kansas State University Institutional Review Board prior to field work. Basic demographic data were also collected. Respondents were asked the same questions in approximately the same sequence, but questions were added, removed, or changed depending on responses. Sessions were digitally recorded (when permitted by the interviewee), transcribed into individual Microsoft Word documents, and analyzed using Atlas.ti, a qualitative data analysis program ideal for coding of texts [85]. Coding and content analyses of

interview data were based on grounded theory, an inductive and systematic approach that suggests that theories can be discovered or formulated through the constant comparison of collected data [86,87]. Coding consisted of reading each interview transcript and determining the primary emphasis of each response, along with any additional concepts or themes that emerged.

## 3. Results

Key informants were asked about the biggest challenge their community faced. A total of 31 responses were collected. Forty-three percent of all the interviewees suggested that tourism and the economy are the biggest challenge. This was followed by community development (17 percent), growth and development (15 percent), and federal policy and regulations (nine percent). Also receiving mention were a collective vision for the community (six percent), environmental protection (six percent), and city versus county issues (two percent) (Table 3). The high proportion of responses related to tourism and the economy is indicative of community dependence on the NPS and USFS for economic viability in gateway communities. The number of responses related to community development and to growth and development is logical for one of the fastest growing regions in the country.

**Table 3.** Key informant responses to the question, "What is the biggest challenge your community faces on its transition toward sustainability?" (percentages rounded).

|  | West Yellowstone | Jackson | Red Lodge | Other | Total Responses | % of All Responses |
|---|---|---|---|---|---|---|
| Tourism & Economy | 10 | 2 | 5 | 6 | 23 | 42.6 |
| Collective Vision | 3 | 0 | 0 | 0 | 3 | 5.6 |
| Growth & Development | 0 | 3 | 3 | 2 | 8 | 14.8 |
| Federal Policy & Regulations | 5 | 0 | 0 | 0 | 5 | 9.3 |
| Environmental Protection | 0 | 3 | 0 | 0 | 3 | 5.6 |
| City vs. County | 0 | 0 | 1 | 0 | 1 | 1.9 |
| Community Development | 4 | 3 | 1 | 1 | 9 | 16.7 |
| Misc. (climate change/energy) | 0 | 1 | 0 | 1 | 2 | 3.7 |

### 3.1. West Yellowstone, Montana

The two primary perceived challenges in West Yellowstone are tourism and a year-round economy (41 percent of responses), and winter access to Yellowstone National Park (18 percent). West Yellowstone receives the highest percentage of winter visitors in the region; none of the respondents from the other two communities listed winter access as a challenge. Jackson has a small number of recreational snowmobilers during the winter months, and snowmobiling is virtually non-existent in Red Lodge. Tied directly to the economy and winter access is the issue of a perceived lack of collective vision (14 percent of responses) in the community. Without a unified collective vision, there was the growing concern that West Yellowstone would falter.

A big concern for long-time business owners was the high turnover rate for new businesses and the potential impacts this might have on community dynamics. As one business owner put it,

> *Unfortunately, part of the problem here is businesses come in...and buy a business for two years. He leaves . . . and you don't have a solid community. There are a lot of people who have been here with a lot of history, they tend to be cliquish with the newcomers and then the newcomers leave, and what are you left with?*

The same informant followed by saying that, while problems exist with business turn-over, some in the community have stepped forward to be more receptive to the needs of new businesses. Specifically, it was noted that the town planner has "tried to bring all factions together", but it has been difficult. Related to these issues is the pace of change taking place in the community and the diverse backgrounds and changing approaches to business operations that newcomers bring with them.

There is also a substantial issue related to an aging population and the implications this might have for community cohesion and a collective vision. A Chamber of Commerce represented noted such difficulties when she asserted that

> . . . *the leaders of the community...are really starting to retire or have passed away. They had a vision of what they thought West Yellowstone was that has not necessarily transposed itself to the newer, younger business owners. And because we've got such different interests, different perspectives, it is extremely hard for this community to agree on what our strengths are.*

The primary concern was how a community can cope with a rapidly changing and dynamic relationship with the national park, while continuing to support the local citizenry and attract tourists. This point was illustrated well by an informant when she said that

> *... by the time we've got done addressing one thing we are already confronted with a new issue. So, you know a perfect example is access to Yellowstone in the winter. You know when you spend years working really hard to build a business in a certain direction and then you have to respond to courts, and then you have to respond back to environmentalists, and then you have to switch. Nowhere else have I seen businesses that have to totally change how they think, operate, and market as much as they do in West Yellowstone.*

One of the largest perceived challenges West Yellowstone faces, and one that "permeates most everything that happens in town, is the age-old challenge that we have been wrestling with is the development of a year-round economy", according to one respondent. How can a community survive on an economy that lasts 100 days at worst, and 120 days at best? Two issues were prevalent in comments related to a year-round economy: a lack of shoulder-season employment opportunities, and the retention of quality employees. As a business owner noted, "the winter season continues to be doing worse . . . for many businesses in town. Over the last three, four or five years, there are fewer and fewer businesses open in the wintertime. We've gone backwards" toward a one-season town.

Several business owners suggested that the best defense against transitioning back to a one-season town is customer service. Related observations are that "it turns out that we have to provide good service to our visitors . . . we can't take these tourists for granted" and "whether we like it or not, [West Yellowstone] is in the business of customer service." A big concern is attracting employees "who are in tune enough, or care enough" to ensure the customers have the best experience possible will not only return themselves, but tell others about their experiences in the community.

When interviews were conducted, West Yellowstone had spent nearly two decades caught between proposed management directives from the National Park Service regarding the number of snowmobiles allowed in Yellowstone National Park during the winter months and environmental groups who continually litigated against such changes due to perceived negative impacts on the natural environment and user experience. Since that time, a new winter use plan was approved by the NPS and went into effect during the 2014–2015 winter season. The plan allows for increased snowmobile activities through communities like West Yellowstone, and has been embraced by locals for being a balanced approach to limiting environmental impacts while enhancing economic benefits [88]. However, the prolonged debate over winter access remains a concern for residents. "There's a long history. Fifteen years we've been going on this snowmobile thing and there's an awful lot of water that has come under the bridge." The unsettled situation regarding snowmobile use in YNP had resulted in not only decreased revenues for the community, but more importantly, had weakened the very fabric of community life. Speaking of the economic impact of snowmobiles, a city official suggested that

> *Ten years ago our winters were almost as good as our summers . . . they were right there and that made things wonderful and kept more motels open year round, more people employed year round. Now that we don't have that half the town or maybe more is closed down in the winter time.*

Replacing the snowmobile with other tourist and recreation-based activities was perceived as being "really hard". The community and businesses have discussed activities such as snowshoeing, hiking trails, photography, and cross-country skiers. While it is believed that "none of those are bad ideas", they just do not "carry the same impact as snowmobiles". While groups like the Greater Yellowstone Coalition suggest that cross-country skiing can save the town and provide it with the economic boost it needs during the winter months, business owners and city officials are not entirely convinced. Both groups perceived a large economic discrepancy between snowmobiles and cross-country skiers. This was illustrated by a city official who said:

*We always say that one good snowmobile family from Minnesota is the equivalent of a whole lot of skiers because those people eat and drink, party and they enjoy themselves, and that's what we want. You know it's nice to say we have some nice skiers who come to town and they are lovely people . . . they bring their children and lovely families. But they are not hanging out at the bars spending money, they are not going downtown.*

The issue has also made it difficult for businesses or families to survive in the fickle tourist economy. "At one time we could—you and your family—move here and maybe not make a great living, but you could work year round and be in this area and make a living year round. But now that's pretty tough." Decreased visitation rates in the winter months have also resulted in a declining year-round population base, which has in turn translated into decreased community services and declining school enrollments. An informant noted, "Well, the problems that come from that, whether you're at church, or you're at school, or you're at the city offices . . . we've gone backwards".

When informants were asked if they felt that the challenges facing West Yellowstone were unique to the community, and not representative of the region as a whole, the resounding answer was yes. The most frequent response was that West Yellowstone and the challenges it faces are unique because of the community's dependency on Yellowstone National Park. "We are the hub, so to speak, of Yellowstone . . . we've achieved the popularity as one of the main gates." The metaphor of closing or removing the tourist-based attractions was used by one NGO representative to suggest the uniqueness of West Yellowstone:

*You could close the Beartooth Highway tomorrow and, while that's a segment of Red Lodge marketing, they could still get away with marketing all the other things there is to do. If you close access to Grand Teton . . . [that] would affect Jackson in the summer, but they still have the ski resorts and an airport and they could literally supply a lot of activities without even requiring the parks. If you closed our access to Yellowstone, it would be a whole different kind of dynamic here.*

The community is also perceived to be unique in that it relies solely on tourism and public lands—especially Yellowstone National Park and NPS management decisions—for its economic viability, as well as providing the infrastructure and service that local residents and 4 million visitors per year depend upon:

*West Yellowstone is also unique in that it is surrounded on all sides by public lands. So there's not a lot of population around to support the infrastructure and consequently, again it's a double-edged sword. We built this place to be a nice place so more people come. There's about 2 million people that go through here and yet there is no population base to support all the police and fire and of that stuff. So we have to rely on tourism and the resort tax. We have to be innovating in these other types of things in order to make a go. There is nobody else like that.*

Although most respondents felt that West Yellowstone was unique, a small group recognized the similarities to other gateway communities in the region, including dealing with the shoulder season and providing goods and services to tourists. However, even these respondents felt that issues like winter access to Yellowstone National Park and the on-going bison and wolf debates make

the challenges the community faces much more pressing and affect their community far more than other communities.

This study suggests that the primary challenges facing the community of West Yellowstone on its transition toward sustainability are federal policies and on-going litigation involving winter access to Yellowstone National Park, the community's dependence on YNP for economic vitality, and the lack of a collective vision for the community's future. Although concern for diversification and a year-round economy permeated all three study communities, it was manifested differently in West Yellowstone than in the other two communities: only West Yellowstone informants perceived winter access to Yellowstone National Park as a challenge to sustainability. All informants recognized detrimental impacts of shifting federal policies and on-going litigation by environmental groups regarding winter access. However, there were widely mixed opinions on how to best move forward. These included beliefs that diversification of the economy through additional tourist industries such as casinos and amusement parks are the solution, that plowing park roads during the winter months would ensure economic vitality, that the fight for snowmobile access to the park should continue, and even that moving outside the region would be in their own interest.

### 3.2. Red Lodge, Montana

In Red Lodge, the largest perceived challenge was tourism and the economy (50 percent of respondents), followed by growth and development (30 percent). Like West Yellowstone, community economics are largely driven by tourist visitation and spending, and because of this, "you see businesses struggle." This is especially true during the winter months. "We need to draw more people regionally to our areas because winter is not that successful of a season." One business owner stated that 60 percent of gross sales for "the businesses downtown, the little shops and restaurants downtown ... happens between Memorial Day and Labor Day." Hence, for 10 weeks local businesses conduct strong business, saving revenues, and then spend "42 [weeks] spending, because the money that comes in winter does not sustain the businesses." Another business owner said that "getting that dollar is Red Lodge's biggest problem."

Like West Yellowstone, it was suggested that one of the other primary issues related to tourism and economics was a lack of collective vision for the community. However, some argued that even with a collective vision, the community would still suffer economically because of its proximity to Billings, the regional economic hub. This was expressed by a business owner who was also actively involved in community planning:

*That is the challenge for communities like Red Lodge when you are so close to the largest business center in the region, which is Billings. You are just not going to redevelop your downtown to where you ... meet all the community needs within the city limits. That is not going to happen. And there are dreamers out there who are not business people, who think it is just a matter of getting the right mix and it will all come together. I go to the meetings. I participate in the discussion and tell them over and over and over again that you are never going to see that happen again in small town Montana.*

Some business owners also felt that large events (e.g., the Beartooth Rally, a well-publicized national motorcyclists' gathering) were benefiting a small portion of the business community at the expense of the larger community. It was suggested that

*Local business owners whose livelihoods depends on the three months of summer and tourism feel, some of them feel, like they lose a week of business during that week because your typical ... bike rally attendees aren't typically there to browse the stores, the souvenir store and things like that. There's so many bikes that come into town that it tends to take up parking so people who might have stopped roll on through. And then street dances—streets are closed for dances and the business owners on those streets feel like their customers don't even have a shot getting into them.*

Growth and housing development are commonly seen as another of the most important challenges to Red Lodge. This topic included three primary issues: supporting the existing quality of life,

the impact of amenity migrants, and the implication of large-scale developers purchasing land around the community. Most current residents were drawn to Red Lodge for its unique quality of life, and not its economic potential [78]. The challenge, however, was perceived to be how to "enjoy what we have while sharing it with others." It was suggested that through sustainable planning, most notably reducing sprawl and protecting scenic and wildland values, this should be possible.

Noting a trend in recent years of increasing numbers of 10 to 20 acre lots, one business owner said, "We need to take those 20 acre ranchettes and make them into small lot subdivisions . . . where you can get as many people into an area . . . and preserve those things that are valuable." "Small town. What's the word . . . small town being" and similar expressions were used to describe the uniqueness and the quality of life in Red Lodge. The problem associated with the associated sentiments was that many residents wanted no growth at all. Responding to the "no-growthers", a city official stated, "They don't understand that growth is inevitable; it's going to happen. We can't stop it, but we have to control it." Controlling growth was seen as a fundamental obstacle for the community, primarily because Carbon County does not have planning or zoning ordinances, while the city of Red Lodge does. As one city official perceived the city vs. county issue, "the city has really worked hard, I believe, in keeping what we have . . . but still I don't believe the county—I don't think they are on top of their game." With a "hands-off" approach at the county level, local residents are "starting to lose what they value the most. Recreational opportunities are disappearing because big companies, big business people are coming in and they are buying up ranches and they are closing off those recreational opportunities."

One business owner felt the problem is a "failure to really look at the consequences of our decision-making process . . . and what the long-term outcome of that decision-making process is." The concern was also that, as a small community, Red Lodge could become" . . . victimized, but it doesn't quite capture what I want to say." That "victimization" would likely come from a large-scale developer, who might take advantage of limitations inherent in small communities. As expressed by a business owner,

> *I think it's easier for a sizeable developer, someone with deep pockets to really come in here and do things in a way that would not have the kind of resistance they would have in a community like Jackson.*

While the community has been reluctant to let large-scale developers "have their way", it was perceived that community members "don't have the ammunition to really stand up." As a result, Red Lodge is "more reactive than proactive." In contrast, some in the community felt that large-scale development can be a good thing because big corporations "are used to working in mountain towns all over the Rocky Mountain West, and their track record is really strong." This group also recognized the potential pitfalls of big corporations having a stake in local development. One of the main fears was that big corporations "might choose to make an investment in Red Lodge because they valued the kind of character of this community. But they are not tied to that character, and they're not invested the same way that people who live, work, and play here are." To help guide the community through the potential pitfalls of rapid and unregulated growth, city officials, and the public at large, look to communities like Bozeman and Jackson for guidance. As one business owner noted,

> *At some levels we are fortunate that those communities have kind of gone before us and we can learn from them and I think that people in our community that are really on, in that kind of realm, are sensitive to that and are trying their hardest to get us out front of that.*

One of the other fundamental concerns from respondents is the influence that recent amenity migrants are having on the community. Of particular concern is a lack of community attachment and involvement by newcomers that often translates into a desire for goods and services that had not traditionally been available in the community. A city official speaking on this issue noted his frustration:

*We don't have a Walmart; we don't want one. You know, you're going to pay more, your quality of food is maybe not going to be as good at the grocery store, we're at the end of the line kind of thing. You are going to pay more for gas. So don't start trying to change it and make it what you left, because pretty soon it's going to be the same thing you left and then it's why did you leave there anyway? If that's what you wanted, why did you leave?*

When asked if the challenges facing Red Lodge are unique to the community, one business owner summarized the feelings of most respondents when he asserted that

*I would say other communities—every community that borders the park—has the same challenge. They are just at different points on that time line or continuum, and on a different scale. And then the political climate is different. Wyoming is different than Montana politically, so what goes on in Cody, for example—our closest gateway community, it's amazing how different we are as communities. We are only 60 miles apart, yet we're very, very different. It's funny people down in Cody look longingly at the things we have going on, and our people look longingly at the things that they have going on, so there's a lot of mutual envy, but yet neither community would say 'I would want to be like that.'*

Others looked to other communities as examples of what not to do in relation to planning and growth. "You know I think they probably face similar problems and unfortunately I think for both of them (Jackson and Bozeman) it's too late. I don't think they did their job," one business owner explained.

For Red Lodge, the primary challenges to transitioning toward sustainability included dependence on a tourism and recreation-based economy, and growth and development. Attractions including the Beartooth Pass and the Beartooth Plateau are the primary tourist draws in Red Lodge, and while cultural events such as the Festival of Nations and the Home of the Champion Rodeo may bring additional tourists to the community during the summer months, the lack of a diversified economy was perceived as a primary challenge. Like West Yellowstone, the lack of a collective vision for the community's future amplifies challenges to a sustainability transition. Recent growth and development were also viewed as a challenge, primarily based on the lack of countywide zoning that had pitted long-time residents against amenity migrants. Although the community had been proactive in limiting growth within city limits, concern existed over how growth in peripheral regions of the county might impact not only the natural environment but also community character and quality of life.

*3.3. Jacson, Wyoming*

The largest perceived challenges to Jackson, based on informant responses, was growth and development (25 percent of all responses), community development (25 percent), and environmental protection (25 percent), followed by tourism and the economy (16 percent). In the minds of most respondents, environmental protection and growth and development issues go hand in hand, as do tourism and the economy. Thus, separation of growth and development and environmental protection into two categories here is mainly a matter of individuals' emphasis in their responses rather than a complete difference in concerns. The connections are largely because the economy is driven by tourists, who are brought to the community for its unique natural setting. The natural environment has also driven development and growth and thus, in order to sustain the local economy and its tourist base, the natural environment must be protected. One method to ensure environmental protection and the maintenance of the community's tourism base was the city's comprehensive plan that was in the process of being re-written at the time of the interviews. Reflecting on this process, a city official declared

*I think that as we worked on the comprehensive plan we have moved away from the idea of a straight balance of the triple bottom line, to the idea that we want the social equity, and we want this strong economy, both of those thing when you look at them are dependent on the ecology of the area. So keeping that open space, keeping the wildlife, it is our economy, it is why people move here, it does bring all the people together so it is our society as well, and I think that is, the one thing I would add to that personally, I think it has to be a bigger, a bigger picture discussion. You can't ship your impacts over the pass down the canyon, and think that they go away and you're protecting the ecosystem, because they are an ecosystem too.*

A town planner, recognizing large open spaces and wildlife habitat, suggested that the location is "unique in the lower 48 (states) that no one else can really say that they've got a lot of what we've got." A city official echoed a question and concern for many when she said, "everybody has a strong connection to this place and we all want to protect it, and we all think it's beautiful. But we also want to live here . . . so how do you find a way?"

Others felt that, while many acknowledge the importance of the natural environment, few have stepped forward to protect it. A local land trust representative suggested that while three million people travel to the community each summer for its wildlife and natural splendor, few in the community have made the "leap that we have to take care of this". A second land trust representative said,

*I think the biggest challenge for Jackson is to find the balance between where do we want our open space and where do we want our density. And the comprehensive plan has run smack dab into the middle of that and it's a train wreck right now. Politically everyone stands up and says, our natural resources, our habitat is the most important thing . . . but it isn't, there is nothing actually promoting open space.*

Some interviewees expressed frustration that city and county officials were not doing their part to protect the uniqueness of the area, with one stating that officials are "nibbling away at those regulations that would protect those resources," and that it is part "lack of awareness on some people, and a lack of willingness politically on others." However, one land trust representative was optimistic that the problem stems from a lack of education and not a deeper, fundamental problem.

A great deal of frustration regarding open space and development stemmed from recent amenity migrants to the area that are perceived at times to have a limited attachment to place. One informant sympathized with amenity migrants when she stated, "you can't fault them for this, but when they come to Jackson they don't necessarily want to become embroiled in the political scene for the six weeks or the two months that they are here. They want to enjoy what's here." However, this same informant was concerned that "as it becomes harder and harder to live here, we're going to see . . . the community value base sort of erode." The concern that living in Jackson for most long term residents is difficult and has potential long-term implications was highlighted in the comments of a city official who said:

*Just the cost of living here is high and . . . it is very, very difficult to come here as a young person and then make a living to the point where you can afford to stay here long-term. And without that progression of people who were coming into the community and investing in it personally with their time and energy, and able to live here, it will suffer a lot in the long term. Because eventually . . . it's just going to degrade to either a tourist only, or a seasonal resident only environment, and that is when I think you really do start to erode your base of people who make it a good place to live, and who have the values that can form those long term themed decisions and get you . . . focusing on more than just ... this season's bottom economic line.*

Although the debate about no growth, slow growth, and smart growth continues in Jackson, many feel that the community and the county are at least partially taking steps to correct perceived errors of the past. A city official speaking on growth said,

*I think there are very few people in Teton County who would tell you they are pro-growth. I don't know that any of us really want this place to get any bigger, but I think . . . what I would strive for is responsible growth, and balanced development. I think we do it such a dis-service to say we don't want to growth and we're not going to plan for it, because we know that is not the way it works.*

When asked if the challenges that face Jackson are unique in the GYE, most responded that they were not. However, most felt that while communities are similar in the challenges that they face, challenges in Jackson are greater because of the rate and extent of recent growth in the community, and because of the economic disparities between Jackson and the other communities. As a city planner suggested, "I think that all the different communities are in a different spot economically and in a different spot in their growth. We do have the luxury ... to already be established and have the wealth." An NGO representative said,

*I think they have the same problem. I think they don't have it to the degree we do. I'm not sure that they have experienced the kind of growth, especially in second home ownership that we have seen. I mean, look at West Yellowstone and it's still pretty tiny. It's doesn't have 10,000 square foot homes, to the extent that we do. I think I would probably guess that they recognize it. And if they had the opportunity to go all out and develop they would do it.*

In Jackson, the primary challenges were associated with growth and development, connected with environmental quality. Perceptions again pitted long-term residents against recent amenity migrants. The rapid and continuing increases in migration to the Teton Valley also resulted in concerns over the impact of such growth on the natural environment, as well as the availability of affordable housing. The challenge was how to maintain the open spaces, large tracts of undeveloped land, and the high quality of life that had driven population growth in recent decades while simultaneously ensuring the continued success of the community's year-round economy. Although the protection of the natural environment has been considered a paramount concern, with language in past and current comprehensive plans directly focused on the environment, there was growing frustration over the lack of implementation of environmentally-oriented policies at the city and county level. Instead, the perception existed that implementation of such policies is more often than not sacrificed for growth and development, and would ultimately result in the lack of community cohesion and the degradation of the natural environment.

## 4. Discussion

In all three study communities, direct drivers of change, including dependence on tourism and recreation-based industries and the lack of diversified economies, and indirect drivers, such as continued growth and development, have resulted in disconnects among perceptions, priorities, and goals as they relate to sustainability. In addition, physical constraints (space and land ownership) and situations related to external (county-level) land use decision-making affect the three communities. Members of each community were focused on multiple challenges that further complicated the fulfillment of sustainability objectives in terms of all three key considerations—economy, society, and environment. Many concerns connect to amenity migration to the region, with effects on the three dimensions of sustainability and place [3]. The multi-challenge orientation of study communities is reflected in the multiple visions that various stakeholders have for their community and their futures. For communities to be sustainable over the long-term, decision makers and stakeholders must share common goals, visions, and values [49]. In doing so, not only will they strengthen the well-being, community cohesion, and trust of local residents, more importantly, they will ensure community resilience to on-going and sudden changes [46]. For the GYE, what is needed most is a hierarchical approach to a sustainability transition, with each community setting its own—and connected ecosystem-based—goals, objectives, and visions. By doing so, communities will not only possess a guide to begin the transition to a sustainable future, but will also ensure the health of the

Greater Yellowstone Ecosystem. Such community and wider scale efforts must be ongoing, with recognition of concerns beyond city boundaries.

The results from this study and previous studies [10,41,89] also indicate that a transition toward sustainability is manifested differently in the Greater Yellowstone Ecosystem, and potentially other communities adjacent to protected lands, than it is in other areas because of its unique milieu. While the close proximity of federal lands, and thus extra-local decision-making, may complicate the sustainability discourse at times, these same federal lands may also favor a transition toward sustainability in amenity-driven gateway communities [41]. Because of the federal oversight of public lands and their mandates for multi-generational protection of their resources for public enjoyment and use, the communities of West Yellowstone, Jackson, and Red Lodge are better able to focus attention on sustainability objectives and goals oriented towards the local economy and community. This is not to say that environmental dimensions of sustainability can or should be neglected in gateway communities or their adjacent private lands. However, the sustainability of the Greater Yellowstone Ecosystem requires that the mosaic of private lands that connect and act as corridors between federal lands must also be maintained in a sustainable manner. While federally protected public lands are considered the centerpiece of the GYE, societal and economic dimensions of sustainability goals and objectives are often focused on private lands and the character of local communities. The recognition that the sustainability of private lands is a high priority was repeatedly seen throughout this study. While the community of West Yellowstone is geographically constrained and virtually unable to grow spatially, most respondents agreed that preserving the character of the community was of utmost importance [90]. For Red Lodge and Jackson, their continued efforts to preserve open space in and adjacent to their town centers is further evidence of the importance of sustaining environmental attributes on private lands [10].

This study has also indicated that West Yellowstone, Red Lodge, and Jackson stakeholders are conscious of the embedded nature of local communities, economies, and natural environments [3], although this is seldom verbalized. The challenge that remains for these communities is the recognition that while collective visions and comprehensive plans are beneficial to each community individually, a broader vision would ensure not only the sustainability of local communities, but the sustainability of the entire ecosystem which is of such importance to them.

## 5. Conclusions

The science of sustainability attempts to improve our understanding of complex human-environment interactions and seeks to find solutions to issues facing a sustainability transition [5,68]. Because the values, attitudes, and behaviors of people affect actions and priorities, it is essential to identify the challenges that communities face on their sustainability transition, as it aids understanding and decision-making [5]. The goal of this study was to understand better the challenges present for transitions toward sustainability in a coupled and complex human-environment system, the Greater Yellowstone Ecosystem. One of the insights gained from this study is that communities that seemingly have much in common due to their shared region, physical environmental surroundings, and economic dependence on tourism and recreation, their specific concerns to ensure place-based sustainability (focused on all three sustainability dimensions; environmental, economic, and societal) can show wide variability. This is consistent with the idea that sustainability transitions are geographic processes [44], and that there will be differing levels of sustainability for any location. Therefore, a successful transition toward sustainability is ultimately about understanding local contexts and the choices made by individuals and communities [3,11,45].

Variations in local context among communities ultimately result in variations in perceptions, priorities, and challenges related to the transition toward sustainability. Analyses of sustainability objectives and challenges at multiple localities and investigation regarding how transitions toward sustainability are affected by local context would allow for a comparative approach across multiple scales and environments. These would inform scholars and decision makers as they address similar

topics in different locations. Such research will prove valuable in advancing our understanding of the complexity and sustainability of coupled nature-society systems. As Kates [5] suggested, the ultimate goal of sustainability science as a discipline is in not only creating knowledge, but in moving that knowledge into action. Thus, in order to facilitate changes in behaviors and actions, future research should focus on local contexts, with a specific aim of identifying proximate and underlying drivers of change, and the individual and structural barriers that promote or hinder sustainability transitions.

**Author Contributions:** Conceptualization, R.D.B. and L.M.B.H.; Methodology R.D.B. and L.M.B.H.; Investigation R.D.B.; Formal Analysis R.B.; Validation R.D.B.; Writing—original draft preparation R.D.B.; writing—review and editing, R.D.B. and L.M.B.H.

**Funding:** This research received no external funding.

**Conflicts of Interest:** The authors declare no conflict of interest.

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
