# Peer review of "Embedded in Nature: Challenges to Sustainability in Communities of the Greater Yellowstone Ecosystem"

_sustainability, doi:10.3390/su11051459_

Round 1
Reviewer 1 Report
I reviewed this article before and the comments were addressed then. My only concern is that the article is too similar to a dissertation that already has been published. I would recommend paraphrasing when applicable and citing the dissertation when applicable. I am attaching a report from ithenthicate where it shows the sentences that are similar to others in the web. I consider these changes just text editing and nothing major.

Author Response
After consulting with the editor, it was determined that we would not re-write/paraphrase content that was previously published in a thesis/dissertation based on journal policy. If there remains concern, please feel free to reach out to the editor for clarification of this policy.
Reviewer 2 Report
Here are a few corrections: Line 22: ...differ based ...
Line 79: based ... Line 48: remove the first comma
Line 210: community's
Line 316: this line starts too far to the right.
Other than these minor corrections, the text reads quite well.
Author Response
The changes suggested by the reviewer were made and the manuscript was proofread an additional time to correct further grammar issues.
Reviewer 3 Report
I appreciate the extensive revisions that were made to this paper. The framing at the beginning helped to place this research within a theoretical construct.
I have one final recommendation. The current discussion seems like each paragraph should be the end paragraph for each case. I view the discussion as a way to reflect on the findings given the literature review/theory. It seems that this sort of discussion starts on line 680 through about 727. This leave a much shorter conclusion, which is fine in my view.
Author Response
Thank you for this suggestion. The Discussion and Conclusion sections were revised accordingly based on reviewer feedback, and it is agreed that this format flows much smoother and provides more clarity.
This manuscript is a resubmission of an earlier submission. The following is a list of the peer review reports and author responses from that submission.
Round 1
Reviewer 1 Report
Thanks for the opportunity to review this interesting paper examining stakeholder perceptions of the sustainability of resource-dependent communities in the GYE region. I believe the topic area is highly relevant and significant to readers of this journal. I found the manuscript to be clearly written and generally well organized. Appropriate secondary and primary sources of data were brought to bear on the research questions addressed. Logical conclusions were reached based on the evidence presented.
I do have some concerns about the manuscript though. One weakness of the manuscript at this stage of development is the lack of any theoretical foundation for the research questions addressed and research design employed and conclusions reached based on that research design. The stated study purpose--to examine the challenges communities face on their transition to towards sustainability--is not couched within the published research on this topic. In fact, nowhere in the introduction, is any research on this topic presented. What research has been done on this topic? What is our current state of knowledge? What gaps in the research does this investigation address? The net result is a very descriptive, a-theoretical manuscript which is fine if this is a consultant's report. However, our goal here is a scholarly article that is moving the state of knowledge forward so I suggest that the author integrate more research-based citations throughout the manuscript. How was "community sustainability" conceptually and operationally defined for the research study? Or, were all study respondents allowed to define this term for themselves? Please clarify.
I also have some concern over the sampling procedures used for the study. The researchers purported to use a "purposeful" sampling approach. However, no sampling criteria were provided for the reader. How was the sample purposeful? What was the objective here? The manuscript states that 32 interviews were conducted with key informants. Also, that basic demographic data was collected. However, no demographic profile of respondents was presented in the manuscript. What industry sectors did they represent? What percent of your sample represented ngos? Federal, state, and local government? Providing this demographic profile of respondents is critical to more deeply understand and interpret the tenor of their responses.
So, to conclude, this is an interesting and relevant and timely manuscript. However, it is presently a very descriptive and applied analysis and needs to be more deeply nested within the empirical research that's been conducted on this topic. Once integrated with relevant research, the manuscript could make a fine contribution to this journal.
Author Response
Thanks for the opportunity to review this interesting paper examining stakeholder perceptions of the sustainability of resource-dependent communities in the GYE region. I believe the topic area is highly relevant and significant to readers of this journal. I found the manuscript to be clearly written and generally well organized. Appropriate secondary and primary sources of data were brought to bear on the research questions addressed. Logical conclusions were reached based on the evidence presented. Point 1: I do have some concerns about the manuscript though. One weakness of the manuscript at this stage of development is the lack of any theoretical foundation for the research questions addressed and research design employed and conclusions reached based on that research design. The stated study purpose--to examine the challenges communities face on their transition to towards sustainability--is not couched within the published research on this topic. In fact, nowhere in the introduction, is any research on this topic presented. What research has been done on this topic? What is our current state of knowledge? What gaps in the research does this investigation address? The net result is a very descriptive, a-theoretical manuscript which is fine if this is a consultant's report. However, our goal here is a scholarly article that is moving the state of knowledge forward so I suggest that the author integrate more research-based citations throughout the manuscript. Response 1: There has been extensive revisions to the introduction and conclusion to include a much more focused sustainability framework, especially as it applies to the concept of sustainability, a transition towards sustainability, and the challenges society faces related to transition. (Lines 42-98, 725-786) Point 2: How was "community sustainability" conceptually and operationally defined for the research study? Or, were all study respondents allowed to define this term for themselves? Please clarify. Response 2: “Community development” was defined (broadley) by individual respondents Because the primary focus of this research was to report on perceived challenges to a sustainability transition by participants, a detailed examination of how respondents define the concept of sustainability was withheld. . Point 3: I also have some concern over the sampling procedures used for the study. The researchers purported to use a "purposeful" sampling approach. However, no sampling criteria were provided for the reader. How was the sample purposeful? What was the objective here? Response 3: Language was added that reinforces the benefits of purposeful sampling and its objectives. (Line 204-205) Point 4: The manuscript states that 32 interviews were conducted with key informants. Also, that basic demographic data was collected. However, no demographic profile of respondents was presented in the manuscript. What industry sectors did they represent? What percent of your sample represented ngos? Federal, state, and local government? Providing this demographic profile of respondents is critical to more deeply understand and interpret the tenor of their responses. Response 4: Added Table 2 - Respondent Demographic Data. (Line 217-218)
Reviewer 2 Report
The authors do not make clear the link between sustainability and their analysis. They need a sustainability framework or some mechanism that shows how the interviews and other descriptive data show or not how the area is or is not moving towards sustainability.
Also, they give % growth, which is a bit deceiving. A small place may look like it's growing a lot in terms of %, but in sheer numbers is growing little - a place that grows by 50% but is only 1,000 at time 0, is only 1500 at time 1. While substantial for that community perhaps not for the entire county. perhaps relative % growth in relation to the county would put growth in context.
Table 1 data is either at the county level or at municipal level but over what time period? Also, small places have a notoriously high error term and should be explained in a footnote and what it is.
Commuting is mentioned but not how far. Is it a two-hour commute or 20 minutes?
On the ethics question, just wondered if this went through a campus IRB approval process.
In discussing various topics, such as mining, a map might be useful especially in relation to the communities.
Really, one of the fastest growing regions? by % or number? What does this really mean?
There is lots of description but this article needs to be more analytical. perhaps this is useful for the communities, but I am not sure it leads the readers of this journal anywhere.
The link with sustainability is weak at best. The analysis needs a framework of sustainability. I don't know if they are talking about - what kind of sustainability. Which aspects of it? Does resilience play a role here? I would like to see a far more analytical article than the report-like article it is now.
Author Response
Point 1: The authors do not make clear the link between sustainability and their analysis. They need a sustainability framework or some mechanism that shows how the interviews and other descriptive data show or not how the area is or is not moving towards sustainability. Response 1: There has been extensive revisions to the introduction and conclusion to include a much more focused sustainability framework, especially as it applies to the concept of sustainability, a transition towards sustainability, and the challenges society faces related to transition. (Lines 42-98, 725-786) Point 2: Also, they give % growth, which is a bit deceiving. A small place may look like it's growing a lot in terms of %, but in sheer numbers is growing little - a place that grows by 50% but is only 1,000 at time 0, is only 1500 at time 1. While substantial for that community perhaps not for the entire county. perhaps relative % growth in relation to the county would put growth in context. Response 2: Reference was added (Line 119), growth rate added (Line 120). Further reference to projected population changes for the region were added (Line 121-122) Point 3: Table 1 data is either at the county level or at municipal level but over what time period? Also, small places have a notoriously high error term and should be explained in a footnote and what it is. Response 3: Date of data collection/analysis was included (Line 178). All data is collected and analyzed at the community level. Point 4: Commuting is mentioned but not how far. Is it a two-hour commute or 20 minutes? Response 4: Distance and height for the Teton pass was added (Line 195-196) Point 5: On the ethics question, just wondered if this went through a campus IRB approval process. Response 5: Language was added to reflect approval from the IRB. (Line 220) Point 6: In discussing various topics, such as mining, a map might be useful especially in relation to the communities. Response 6: Unfortunately, because of the numerous administrative agencies that are responsible for this region, there does not exist a unified (or even modestly accessible) source for such data. Each state has separate data for each subject (i.e. mining), which would need to be aggregated and then separated into individual mining, oil, gas, etc. types. It is believed that such work may be overly burdensome for the scope of this project. If reviewers know of more accessible data for the region the author would be more than happy to pursue such options. Point 7: Really, one of the fastest growing regions? by % or number? What does this really mean? Response 7: Reference as added (Line 119), growth rate added (Line 120) Point 8: There is lots of description but this article needs to be more analytical. perhaps this is useful for the communities, but I am not sure it leads the readers of this journal anywhere. Response 8: Please see revisions made to the introduction and conclusions as efforts were made to move away from the more applied nature of this work. Point 9: The link with sustainability is weak at best. The analysis needs a framework of sustainability. I don't know if they are talking about - what kind of sustainability. Which aspects of it? Does resilience play a role here? I would like to see a far more analytical article than the report-like article it is now. Response 9: There has been extensive revisions to include a much more focused sustainability framework, especially as it applies to the concept of sustainability, a transition towards sustainability, and the challenges society faces related to transition. (Lines 42-98)(Lines 725-786)
Reviewer 3 Report
This is a very good article, based on a very interesting methodology that is very much based on interviews with different actors.
There are a few (about a dozen) minor corrections to the text necessary (e.g. punctuation, and few changes to words (e.g. line 199, change 'who' to 'which'.
It is certainly fascinating and very pertinent regarding 'sustainability'.
The only point that I would make is that a form of strategic development planning might be very useful to this type of territory, e.g. strategic development planning FOR and BY the citizens. There are many examples of this in different parts of North America and the results have generally been extremely constructive and recognized as such by governments at all levels.
Author Response
This is a very good article, based on a very interesting methodology that is very much based on interviews with different actors. Point 1: There are a few (about a dozen) minor corrections to the text necessary (e.g. punctuation, and few changes to words (e.g. line 199, change 'who' to 'which'. Response 1: Manuscript was vetted for additional grammar and punctuation errors. No specific lines were included. It is certainly fascinating and very pertinent regarding 'sustainability'. Point 2: The only point that I would make is that a form of strategic development planning might be very useful to this type of territory, e.g. strategic development planning FOR and BY the citizens. There are many examples of this in different parts of North America and the results have generally been extremely constructive and recognized as such by governments at all levels. Response 2: I appreciate the suggestion for the potential to include strategic community and development planning, with a specific eye for citizen engagement and participation. It is my contention that this falls outside the purview for the manuscript in question here, although it is certainly a point I will take forward as I re-engage with these communities in focus groups and workshops.
Reviewer 4 Report
· I really like this paper. The subject is fascinating, and the paper is well-written! Your findings are both compelling and useful. I learned a lot about this important subject.
· After reading the whole paper, especially the results section, I feel the intro really needs to speak about what we are about to read. Tell me some of your findings at the beginning.
· Define sustainability from the start. It seems the authors uses a definition of the triple bottom line, but this could be made clearer. But then in the conclusion, it seems you were referring to sustainability just from the environmental perspective?
· Would it be helpful to state why previous studies downplayed the value of sustainability from a social perspective? Even when you talk about the tourist economy, you seem to be worry about it from the social perspective. However, this is not made entirely clear.
· Define ecosystem in this case, is it just natural or it also includes build environment aspects. It seems it does base on the way the arguments are made.
· Give more demographic details but the fastest growing cities in the nation [12]. For example, it could be that 100 people live there and it 25 people move in then the percentage would be really great. Still, as a whole this could be kind of insignificant. Say in numbers what the actual changes were. Same with the following sentence. I do not have e context for the place to understand what is a “350 percent increase”. More background on the population, etc. would give me more context.
· Sometimes you use the oxford comma and sometimes you do not. Try to be consistent.
· Define ‘gateway’ communities and cite literature on this in the lit review.
· You map should highlight Jackson, Red Lodge and West Yellowstone clearer. Instead of having the reader to do the work. Maybe big stars or some other kind of marker would do it.
· I am not sure why these communities were chosen besides they are different. Give more background.
· What are communities? Are they towns? Villages? Neighborhoods? A group of people that share culture like Native Americans? Reservations? Unincorporated areas? I am not familiar with the area.
· It seems that affordable housing is a major concern, you should make this more obvious in the abstract or introduction, outside of the quote you use about the resident and the politicians talking about it.
· Clearly state your research questions.
· Be clear about how drives the policy making of mining, oil, gas, etc. (sections 2.3). Policy decisions that affect these industries are done at the city, county or regional scale. How this relate to the stakeholders that you interview?
· What is the relationship between regional or county issues and the community? Also, when interviewees talked about their community did they also refer to county or regional issues?
· Could cite Rittel for Wicked Problems (1973). He has gotten the most citations.
· You talk about the town planner several times. Maybe more planning readings should be cited?
· It seems as you are using different definitions of community and they are getting all mixed. Maybe you should not label Jackson, Red Lodge and West Yellowstone as communities but more as small towns, or whatever they are. Then in the result section you can talk about the different communities (or interest groups) that exist within the same town (young new business owners vs. the older adults that have lived in the area for a while. Otherwise, the geographical and the social is getting all intertwine in a way that it doesn’t make much sense. This will allow you to be clearer and more specific.
· The survey tables could be better discussed in the text. Are the differences significant, could you compare groups statistically speaking.
· I struggle with the results, maybe it could be organized better?
· I would recommend additional readings:
o Guan, C., Wen, X., Gong, Y., Liang, Y., and Wang, Z. (2014). Family environment and depression: A population-based analysis of gender differences in Rural China. Journal of Family Issues. 35: 481 – 500.
o Kaplan, R. & Kaplan, S. (1989). The experience of nature. Cambridge, MA: Cambridge University Press.
o Konijnendijk, C. (2008). The Forest and the City: The cultural landscape of urban woodland. Springer Science & Business Media.
o McGranahan, D. (1999). Natural amenities drive rural population change (No. 33955). United States Department of Agriculture, Economic Research Service.
o McGranahan, D. (2008). Landscape influence on recent rural migration in the US. Landscape and Urban Planning. 85(3): 228-240.
o Wu, J. and Gopinath, M. (2008). What causes spatial variations in economic development in the United States? American Journal of Agricultural Economics. 90(2): 392-408.
Author Response
I really like this paper. The subject is fascinating, and the paper is well-written! Your findings are both compelling and useful. I learned a lot about this important subject.
Point 1: After reading the whole paper, especially the results section, I feel the intro really needs to speak about what we are about to read. Tell me some of your findings at the beginning.
Response 1: Summary of findings was added (Line 141-145)
Point 2: Define sustainability from the start. It seems the authors uses a definition of the triple bottom line, but this could be made clearer. But then in the conclusion, it seems you were referring to sustainability just from the environmental perspective?
Response 2: A definition for sustainability, based on Clark and Dickson, 2003, was added (line 42-43). More inclusive language was added to the first paragraph of the conclusion that lets the reader know that all three dimensions of sustainability (economic, environment, society) are included. (Line 725-736).
Point 3: Would it be helpful to state why previous studies downplayed the value of sustainability from a social perspective? Even when you talk about the tourist economy, you seem to be worry about it from the social perspective. However, this is not made entirely clear.
Response 3: There has been extensive revisions to the introduction and conclusion to include a much more focused sustainability framework, especially as it applies to the concept of sustainability, a transition towards sustainability, and the challenges society faces related to transition. (Lines 42-98 and 725-786)
Point 4: Define ecosystem in this case, is it just natural or it also includes build environment aspects. It seems it does base on the way the arguments are made.
Response 4: Definition was added to bring clarification (Line 99-100)
Point 5: Give more demographic details but the fastest growing cities in the nation [12]. For example, it could be that 100 people live there and it 25 people move in then the percentage would be really great. Still, as a whole this could be kind of insignificant. Say in numbers what the actual changes were. Same with the following sentence. I do not have e context for the place to understand what is a “350 percent increase”. More background on the population, etc. would give me more context.
Response 5: Reference as added (Line 119), growth rate added (Line 120). Reference to 350% was removed as only this value (not original numbers) was available via citation. Further reference to projected population changes for the region were added (Line 120-123)
Point 6: Sometimes you use the oxford comma and sometimes you do not. Try to be consistent.
Response 6: Manuscript was vetted for consistency in grammar and punctuation
Point 7: Define ‘gateway’ communities and cite literature on this in the lit review.
Response 6: A definition and proper citation was added (Line 161-162)
Point 7: You map should highlight Jackson, Red Lodge and West Yellowstone clearer. Instead of having the reader to do the work. Maybe big stars or some other kind of marker would do it.
Response 7: Map symbology was modified for better readability
Point 8: I am not sure why these communities were chosen besides they are different. Give more background.
Response 8: Study communities were chosen because they are “gateway” communities to Yellowstone and Grand Teton national park, they display differing socio-economic characteristics, and because they possess differing natural amenities that make them desirable as tourist/recreation, and amenity migrant destinations (Lines 157-162).
Point 9: What are communities? Are they towns? Villages? Neighborhoods? A group of people that share culture like Native Americans? Reservations? Unincorporated areas? I am not familiar with the area.
Response 9: Added reference to study communities being City’s or Towns (Line 158, and throughout changes were made for clarify). A specific definition was given on Line 136.
Point 10: It seems that affordable housing is a major concern, you should make this more obvious in the abstract or introduction, outside of the quote you use about the resident and the politicians talking about it.
Response 10: Affordable housing was added as a primary challenge in the introduction (Line 103-109)
Point 11: Clearly state your research questions.
Response 11: Research question was added (Line 139-141)
Point 12: Be clear about how drives the policy making of mining, oil, gas, etc. (sections 2.3). Policy decisions that affect these industries are done at the city, county or regional scale. How this relate to the stakeholders that you interview?
Response 12: A sentence was added that suggests that multiple agencies and institutions are ultimately response for decision making within the region, and this may help or hinder a sustainability transition (Line 246-249)
Point 13: What is the relationship between regional or county issues and the community? Also, when interviewees talked about their community did they also refer to county or regional issues?
Response 13: All references to county or city concerns are explicitly stated - that is, informants did not confound the two. This is shown for all three study community, but is more explicit in Jackson and Red Lodge where there was perceived issues between county and city planning objectives and goals.)
·
Point 14: Could cite Rittel for Wicked Problems (1973). He has gotten the most citations.
Response 14: Reference Added (Line 241)
Point 15: It seems as you are using different definitions of community and they are getting all mixed. Maybe you should not label Jackson, Red Lodge and West Yellowstone as communities but more as small towns, or whatever they are. Then in the result section you can talk about the different communities (or interest groups) that exist within the same town (young new business owners vs. the older adults that have lived in the area for a while. Otherwise, the geographical and the social is getting all intertwine in a way that it doesn’t make much sense. This will allow you to be clearer and more specific.
Response 15: City/Town classification was added throughout
·
Point 16: The survey tables could be better discussed in the text. Are the differences significant, could you compare groups statistically speaking.
Response 16: Statistical analyses were beyond the scope of this project, and are often difficult given the small sample size.
Point 17: I struggle with the results, maybe it could be organized better?
Response 17: After considerable thought, I have decided that the organization of the conclusions beset highlights the individual and collective challenges these communities face on their transition toward sustainability.
Point 18: I would recommend additional readings:
Response 19: The additional readings suggested by the reviewers were examined and included if appropriate.
Round 2
Reviewer 1 Report
i have the same concerns as expressed in the first submission.
Reviewer 2 Report
Thank you for the revisions. It helped frame the case studies you discussed. I would love to see what the possible actions are for these communities. While there are barriers and issues, it's difficult to see how they might be resolved.
I suggest doing a thorough proofreading. There are some grammar and punctuation that needs to get cleaned up.
Reviewer 4 Report
This article is too similar to a dissertation already published. The author needs to rephrase what is being said.
